# Improvements in Swim Skills in Children with Autism Spectrum Disorder Following a 5-Day Adapted Learn-To-Swim Program (iCan Swim)

**DOI:** 10.3390/jcm10235557

**Published:** 2021-11-26

**Authors:** Emily E. Munn, Lisa Ruby, Melissa M. Pangelinan

**Affiliations:** 1School of Kinesiology, College of Education, Auburn University, Auburn, AL 36849, USA; mgp0020@auburn.edu; 2iCan Shine, Paoli, PA 19301, USA; lisa@icanshine.org

**Keywords:** adaptive aquatics, behavior supports, community program, skill assessment, swim lessons

## Abstract

Drowning is one of the leading causes of death in children and teenagers. Individuals with autism spectrum disorder (ASD) are at increased risk for drowning. Improvements in swim skills have been observed in children with ASD participating in learn-to-swim programs. However, it is unclear if age, co-occurring conditions, and/or the dose of practice influence swim skills in this population. To this end, a secondary data analysis of iCan Swim program data was conducted to determine the efficacy of the 5-day adapted learn-to-swim program for a cohort of children with ASD ages 3–16 years (*n* = 86). Participant swim level was evaluated at the start and end of the program. Linear mixed-effects regression was used to examine the effects of Time (start/end), Age, Dose of Swim Practice (i.e., total time–time out of the water), and ADHD status on the overall swim level. Participants significantly increased the swim level from the beginning to the end of the program (B = 0.63, 95% CI = 0.52–0.74), and participants with ASD and co-occurring ADHD had greater swim levels regardless of Time than those without ADHD (B = 0.45, 95% CI = 0.05–0.84). Overall, iCan Swim is effective in improving the swim skills of children and teenagers with ASD.

## 1. Introduction

Drowning is one of the leading causes of death in children and teenagers [1]. In 2019, the American Academy of Pediatrics reprioritized water safety and updated the recommendation to prevent drowning by increasing water competency in high-risk groups [2]. Individuals with autism spectrum disorder (ASD) are a high-risk group with an increased risk for drowning [3]. Moreover, the number of children diagnosed with ASD has increased significantly over the last 10 years [4,5].

Individuals with ASD exhibit lower physical activity levels than their peers [6,7,8]. This may be due to individuals with ASD experiencing barriers to common daily tasks, including recreational and fitness activities. These barriers are often environmental (e.g., lack of programming, safety, untrained staff) [9,10] and social/emotional (e.g., behavioral difficulties, poor social skills, negative stereotyping) [9,10,11]. In addition to the environmental and social/emotional barriers to physical activities, individuals with ASD often exhibit deficits in motor skills [12,13,14,15]. The National Autism Center suggests that physical activity interventions may help in improving motor skills and identifies these interventions as an emerging treatment for the associated symptoms [16]. Therefore, individuals with ASD may benefit from swim intervention programs to help in developing their swim skills. 

Not only do learn-to-swim programs promote lifesaving skills, but they also create the path to access swimming as a recreation and sport, which is an excellent source of physical activity and exercise across the lifespan. Furthermore, swimming as a nontraditional team sport, affords individuals with ASD opportunities for social interactions while performing in either individual or team competition. Beyond Special Olympics Swimming, organizations such as USA Swimming and the United States Masters Swimming both have statements of inclusion, promoting the participation of individuals with ASD in competition. 

Despite these opportunities for participation and competition, environmental (e.g., lack of programming, safety, untrained staff) [9,10] and social/emotional barriers (e.g., behavioral difficulties, poor social skills, negative stereotyping) [9,10,11] prevent individuals with ASD from participating in learn-to-swim classes with their typically developing peers. In addition, there is a lack of specific programming for children with disabilities, which includes ASD. Therefore, there is a need for adapted learn-to-swim programs developed to increase swim competency and reduce the risk of drowning in this population. 

Several studies have shown an improvement in swim skills in individuals with ASD participating in adapted learn-to-swim programs [17,18,19,20,21,22,23]. Each program implemented different methods for teaching these individuals (e.g., sensory exposure, aquatics skills checklist, Humphries’ Assessment of Aquatic Readiness, Halliwick’s Method), program length (i.e., 4–96 h), and number of participants (i.e., 3–30). Many of these programs employed a one-on-one or 2:1 lesson. For example, Lawson and Little used “sensory-focused” swim lessons to teach children with ASD to swim in one-on-one settings [24]. The authors reported an improvement in overall swim skills and sleep quality following 4 h of lessons. Rogers et al. reported significant improvements in swim skills after 30 h of one-on-one swim lessons [22]. Research on the efficacy of group swim lessons with an individual with ASD is lacking. However, this is likely due to the fact that individual instruction is common when teaching individuals with disabilities. Pan studied the effects of a 2:1 swimmer on instructor lesson design. This study showed that significant improvements were made after 14 h of swim lessons with many swimmers reaching independence in the water [21]. A follow-up study was completed with siblings in place of the typically developing peers, using the same lesson design (i.e., 2:1 swimmer to instructor). The researchers once again saw that after 32 h of swim lessons, significant improvements in swim skills were observed for both groups (siblings and individuals with ASD) [19]. Despite heterogeneity in the implementation of adapted learn-to-swim programs, collectively, these results suggest that small or individual programs lead to improved swim skills in children with ASD. 

Group lessons may be an effective means of increasing program capacity. However, there is limited research examining the efficacy of group swim programs for individuals with ASD. Indeed, the feasibility and effectiveness of group lessons on swim skill improvements for individuals with ASD, in particular, are unknown. Therefore, this study examined the iCan Swim learn-to-swim program’s efficacy on the development of swim skills in children and adolescents with ASD. Additionally, this study examined the effects of individual factors (i.e., Age, ADHD diagnosis, and Dose) on swim level. 

## 2. Materials and Methods

### 2.1. Program

A secondary data analysis of anonymized program data from nine iCan Shine swim programs was conducted across the United States from April–August 2019. The program iCan Shine is a 501(c)(3) that works toward inclusion for individuals with developmental disabilities in biking, swimming, and dancing. In addition, iCan Shine hosts iCan Swim camps across the United States each year. The program was implemented by a trained adaptive aquatics swim instructor on 5 consecutive days (Monday–Friday), 45–60 min each day. All of the data were acquired by the instructors and program director to track the progress of each participant during the program. This information was shared with parents and volunteers during the program and was used for setting goals for each session. These data were not previously evaluated statistically for research purposes. Detailed information on the iCan Swim program including the training for volunteers and swim instructors is found at: https://icanshine.org/program-hosts/ican-swim-hosts/ (accessed on 22 November 2021).

Briefly, as part of the iCan Swim program, the hosts recruit at least one volunteer per swimmer to register for the program. Volunteers must: Be 15 years or older, be comfortable in water, attend a 30-min Parent and Volunteer Orientation session, and attend a 90-min Volunteer Training (of which 60-min is in the pool). During the orientation and training, volunteers learn how the iCan Swim program operates, the expectations of volunteers, and how to work with swimmers during the program. Volunteers are assigned to work with each swimmer in order to help them learn basic swim skills (entry, exit, breathing, floating, strokes, kicking, and body position). They provide encouragement and physical support (physical holds, using a pool noodle, using other supports) as needed. Volunteers may be local community members, family members, local swim instructors or lifeguards, etc. Fifteen minutes before each session begins, the iCan Swim instructors and swim instructors in training meet with the volunteers to debrief from the previous session and provide additional, individualized lesson plans for that session. For our study, the swimmers were divided into small groups with a ratio of six swimmers per two instructors. The instructors were certified Red Cross Water Safety Instructors and received training by the iCan Shine staff on implementing adaptive swim lessons for individuals with disabilities. With respect to the instructors, the iCan Swim program has two delivery models: One in which iCan Swim provides the adapted instructors and one in which iCan Swim provides the adapted aquatic instructor training (a minimum of 20.5-h) to implement the adapted curriculum for certified learn-to-swim instructors. 

The adapted aquatic instructor training includes: An introduction to adapted aquatics, basics of adapted aquatic skills, creating an effective class, strategies for success working with individuals with disabilities, specific techniques by diagnosis, behavior management strategies, equipment use, safety considerations, effective communication, managing volunteers, identifying and meeting swimmer goals, tracking progress, planning sessions, overcoming common obstacles, and implementing adapted aquatics in the community. Instructors must complete 4 h of classroom instruction and a minimum of 16.5 h in the pool instruction working with participants.

The iCan Swim program consists of 5 consecutive days with each session lasting 45-min (3–7 years old) or 60-min (8 years and older). Each session has a minimum of three swimmers. To maximize capacity and instructor training opportunities, the program is generally run Monday–Friday from 8:30 a.m.–4:30 p.m. with 30-min breaks between sessions and a 60-min lunch break. A maximum of five sessions may be run daily. The lessons consisted of a group game/warm-up (all groups together), skill practice (small groups), group game, and cheer (all groups together). Participants receive instruction on safe entry/exit, breath control (e.g., blowing bubbles, bobs, forward breathing, side breathing), strokes (freestyle, elementary backstroke; breaststroke and backstroke are included for intermediate/advanced swimmers), floating/gliding (front/back), rolling (back-to-front and front-to-back). Each skill progressed from full support from a buddy, then moved to independent skills on a pool noodle if needed, and finally full independence without support. Swimmers were taught safety skills, including identifying a lifeguard, holding onto the wall for support, and using a lifejacket. Behavioral supports included picture schedules, visual supports, and token systems, which were employed when needed by the swimmer. Additionally, instructors used skill development tools and skill modeling, including hand-over-hand instruction and out-of-water skill practice. Equipment aids were used when swimmers required additional assistance with a concept or skill. For example, to help swimmers practice blowing bubbles, participants were instructed to blow a plastic ping pong ball across the top of the water. 

Skill assessments were conducted on Monday, Thursday, and Friday by the group instructors to determine an improvement in key skills (see the assessment below for additional details). For this study, the overall swim level on the first and last day of the program was examined. 

### 2.2. Participants 

The Institutional Review Board at Auburn University approved the exempt status of this secondary data analysis of anonymized data from the iCan Swim program (IRB#21-555 EX 2111). Therefore, no consent/assent forms were required. As part of the iCan Swim program intake forms, the parents report the primary diagnosis, which was used to identify participants with ASD.

The iCan Swim program includes any individual ages 3 and older with a disability (e.g., ASD, Down syndrome, cerebral palsy, etc.) of all abilities. If a participant has a G-tube, the stoma must be at least 2 months old. Participants with a tracheostomy are not eligible, as it is not possible to guarantee a splash-free environment. Participants with behavior problems are not excluded from the program. However, if their behavior is harmful to themselves or others, they may be asked to leave the program.

For the purpose of this secondary data analysis, participants were included if parents reported a diagnosis of ASD, had complete data for the first and last day of the assessment, and were between the ages of 3–18 years. Note: Four children with ASD did not complete the assessments from the first or last day of the program and were not included in the present analysis. Although iCan Swim does not restrict the upper age range of swimmers, only two swimmers were older than 18 years in the present data. Given the present interest in children with ASD, these two participants were excluded from the analysis.

A total of 86 individuals with ASD (75 male/11 female), ages 3–16 years (M = 8.84 years) were included in the study. A parent-reported secondary/additional diagnosis was used to identify swimmers with attention-deficit/hyperactivity disorder (ADHD; *n* = 17, 16 male/1 female). Additional details regarding ASD or ADHD diagnosis were not provided (e.g., ASD level, age at diagnosis, diagnosing clinician, ADHD subtype). In addition to ADHD (*n* = 17), parents reported the presence of the following as “secondary diagnoses”: Anxiety (*n* = 3), apraxia (*n* = 3), cerebral palsy (*n* = 1), epilepsy (*n* = 2), global delay (*n* = 2), hearing impairment (*n* = 1), intellectual disability (*n* = 1), language delay (*n* = 6), non-verbal (*n* = 5), obsessive compulsive disorder (*n* = 1), oppositional defiant disorder (*n* = 2), sensory processing disorder (*n* = 4), and speech impairment (*n* = 3).

### 2.3. Assessment

The iCan Swim assessment was developed to eliminate bias in the type of swim skills evaluated, indicating that any style of swimming (i.e., freestyle, backstroke, etc.) could be used. The swimmers were leveled into one of four categories at the beginning of the program. Non-swimmers (NS) needed assistance at all times, could not complete or refused to attempt a back float or front float. This group is typically fearful of the water. Beginner-swimmers (BS) needed some assistance, could complete a back float or front float with the help of a noodle or the instructor. This group could hold on to a noodle and perform skills independently. Intermediate swimmers (IS) could swim two body lengths independently using any stroke. This group could also swim back to the wall independently. Advance-swimmers (AS) could swim independently for more than two body lengths, take a breath on their own, and learn additional swim strokes. The swimmers could then use any kind of stroke or technique to advance to the next level. Assessments were completed on Monday, Thursday, and Friday of the 5-day program. Data from Monday and Friday were examined in the present analysis.

In addition, for each of the non-swimmers, the instructors provided details at the end of the 5-day program as to why the swimmer did not progress to the beginner level. The instructor selected one or more of the following reasons: The swimmer needed more time (*n* = 22), the swimmer was fearful (*n* = 16), the swimmer’s behavior stopped their progress (*n* = 2), the camp was stopped early due to the weather (*n* = 2) or the buddy prevented the success of the swimmer (*n* = 1). 

### 2.4. Statistical Analysis 

The model selection procedure was based on a forward-selection process. The primary dependent variable was the swim level. First, the random effects were modeled with a random intercept, then a random slope, and then both the random intercept and random slope. Each model was compared to determine the best fit based on the AIC, BIC, and Log Likelihood. The best fit random effect model only included the random intercept. Next, model selection examined the effects of the between-subject fixed effects (Age, Time Out of the Water, ADHD status (yes/no)), and within-subjects fixed effect (Time (first/last session)). These fixed effects and interactions amongst the fixed effects were added iteratively to the model and each model was tested to determine the best fit model based on the AIC, BIC, and Log Likelihood. The best fit model included Age, Time Out of the Water, ADHD status, and Time. There were no significant interactions amongst these variables in the final model. 

All of the statistical analyses were conducted using MATLAB, version R2018a (MathWorks Inc., Natick, MA, USA). Linear mixed-effects regressions (LMER) were conducted to evaluate the factors that influenced the swim level. Age, Time Out of the Water (a proxy for a dose of practice), and ADHD status were between-subject fixed effects. Time was a within-subject fixed effect. A random intercept was modeled for each participant. Follow-up *T*-tests were used to evaluate differences between groups. The level of significance was set to *p* < 0.05 for all of the analyses. A sensitivity analysis was conducted for each significant fixed effect factor based on the standard error of the observed coefficients in the linear mixed effects regressions, degrees of freedom, and 80% power based on the procedure described in Judd, McClelland, and Ryan (2017) [25].

## 3. Results

The mean start and end levels were m = 0.63 (SD = 0.72) and m = 1.17 (SD = 0.91), respectively. The mean Time Out of the Water was m = 14.77 (SD = 33.7718). Figure 1 depicts the end swim level for each participant based on their starting category (i.e., non-swimmer, beginner swimmer, and intermediate swimmer). Of the 42 participants that started off as non-swimmers (i.e., needs assistance at all times and would not attempt front/back float), 22 increased to the beginner swimmer level (i.e., can complete front/back float independently on a pool noodle). The primary reasons provided for the non-swimmers that did not advance (*n* = 20) were: The swimmer needs more time (*n* = 2), the swimmer was fearful (*n* = 13), and no reason was provided (*n* = 5). There was a significant difference in the amount of Time Out of the Water for the group that remained as non-swimmers, compared to those that progressed to the beginner level (*T*(40) = −2.74, *p* < 0.01), but there was no difference in age between the two groups (*p* > 0.05). 

Of the 36 participants that started off as beginner swimmers, 17 increased to the intermediate swimmer level (i.e., can swim two body-lengths independently with any stroke), and one increased to the advanced swimmer level (i.e., can swim independently greater than two body-lengths while taking a breath on their own with any stroke). Of the six participants that started off as intermediate swimmers, all of them increased to the advanced swimmer level. 

Linear mixed-effects regressions (LMER) were conducted to evaluate the factors that influence swim level. The model was significant and accounted for 83% of the variance in swim level (F(8, 164) = 185.12, *p* < 0.001). Table 1 presents the LMER regression estimates, standard errors, 95% confidence intervals, *T*-statistic, degrees of freedom, and *p*-values for each factor examined. A significant main effect of Time (*p* < 0.001) and ADHD status (*p* = 0.03) were observed. Overall, participants significantly increased their swim level from the start to the end of the program (Figure 2, left). 

Participants with ADHD exhibited a higher average swim level than those without ADHD regardless of time (Figure 2, right). Only four of the 17 participants with ADHD started as non-swimmers, compared with 38 out of 69 without ADHD. An examination of the primary reasons provided for not advancing in level for the participants with ADHD included: The swimmer needs more time (*n* = 4), the swimmer was fearful (*n* = 2), and no reason was provided (*n* = 11). In comparison, for the group without ADHD, the primary reasons provided for not advancing in level included the swimmer needs more time (*n* = 11), the swimmer was fearful (*n* = 13), and no explanation was provided (*n* = 45). Although the ADHD status did not interact with Time Out of the Water or Age, the participants with ASD and ADHD spent slightly less time out of the water than those without ADHD (*T*(84) = 1.96, *p* = 0.05), but were not different with respect to age (*p* > 0.05). 

In addition, a significant Time Out of the Water x Time interaction was observed (*p* < 0.001). Figure 3 depicts the change in swim level (from start to end) with respect to the amount of time out of the water during the program. The size of the circle indicates the number of participants with the same change in swim level and time out of the water. The best-fit regression line is also shown. Participants with less time out of the water exhibited a greater change from the start to the end of the program compared to those with a greater time out of the water. Age did not significantly affect the swim level and did not interact with the other factors (*p* > 0.05 for all).

For the time sensitivity analysis, given 80% power and 166 degrees of freedom, the critical beta would need to be 0.119. Given that the observed beta was 0.628, we had sufficient power to detect this effect. For the ADHD sensitivity analysis, the critical beta would need to be 0.503. The observed beta was 0.445, thus we were slightly underpowered for this analysis. This was likely due to a small proportion of participants with ADHD (17/86). For the Time x Time Out of the Water analysis, the critical beta would need to be −0.003. Given that the observed beta was −0.005, we had sufficient power to detect this effect.

## 4. Discussion

This study aimed to examine the efficacy of the iCan Swim group learn-to-swim program in children and adolescents with ASD. We further examined individual factors that influenced swim levels. Consistent with previous studies examining adapted swim lessons for individuals with ASD with either one-on-one or two-to-one designs [17,18,20,22,23], participants in the present study improved their swim skills during a 5-day adapted swim program. This suggests that group swim lessons are a viable option for individuals with ASD. Previous studies have examined swim lessons ranging from 4–96 h [17,18,20,22,23]. The present results suggest that significant changes in swim skills may be obtained in a much shorter time frame than previously suggested. 

It is possible that the success of this program is due to the use of behavioral supports, as well as the quasi-individualized programming provided by the “buddy” in the water, along with the trained adapted swim instructors. It is also possible that the blocked practice (5-days in a row) helped in maintaining motor skill gains for each day of the program. In comparison, Yilmaz et al. met for 10 weeks (3x/week) [23], and the Ennis project met for 10 weeks (1x/week) [20]. It is unclear if other swim programs would see similar results with a different program formatting (e.g., 3 days, 5 days, 10 days, etc.). Taken together, these results are very promising as the iCan Swim model may be scaled for other learn-to-swim programs. Specifically, the combination of the group structure, well-trained instructors and volunteers, and easy-to-use behavior supports could be easily scalable to many environments. 

Interestingly, age was not a significant factor in the development of swim skills. A previous study by Anderson and Rodriguez found that older typically developing children (i.e., 5- to 7-year-olds) develop swim skills at a faster rate than younger typically developing children (i.e., 3- to 4-year-olds) [26]. Although it was expected that older children with ASD would advance in swim level faster than younger children in the present study, this was not the case. One possible reason for this is the covariance between age and time out of the water, as younger children tended to spend more time out of the water than older children. Therefore, the dose of practice and possibly behavioral difficulties, and not age itself, may be more important factors for the development of swim skills in children with ASD. 

Indeed, when evaluating the reasons provided by swim instructors for children not progressing from the non-swimmer category, most of the children were reported to “need more time” or were fearful. Therefore, a greater dose of practice (i.e., more lessons) may be needed to compensate for time out of the water and enable more time for the children to decrease fear/anxiety to progress in their swim skills. Future programs should provide similar details to determine why children spend time out of the water and do not progress in their abilities.

A quantitative comparison between the two groups found that the groups with ADHD spent significantly less time out of the water than those without ADHD. Participants with ADHD exhibited a higher swim level than those without ADHD regardless of time. In addition, a qualitative comparison between the two groups revealed a smaller percentage of non-swimmers in the group with ADHD (24%) compared to those without ADHD (55%). Moreover, the percentage of children with ADHD that were reported as having fear as a primary reason for not progressing in swim skills (12%) was lower than those without ADHD (19%). Therefore, it appears that the group with ADHD may have previous experience participating in swim programs, which may have resulted in less time out of the water and less fear compared to those without ADHD.

While over half of the participants improved their swim skills in 5 days, only eight of the participants reached the highest level of swim proficiency measured. This means that many participants will need additional swim lessons in order to continue to gain the skills to be proficient and independently safe swimmers. The participants in this study did not disclose previous swim lesson experience. To this note, however, Caputo et al. [18] showed that after 96 h of water safety instruction, 78% of swimmers gained independence in the water [18]. This indicates that a larger dose is needed and is effective for full swim proficiency. It is difficult to compare this research to other swim studies due to all the different assessments used in the literature. The use of a consistent swim assessment, such as the ICan Swim assessment or the Modified Texas Woman’s Aquatics assessment, recommend by Lepore, Columna, and Litzner [27], would allow for a better understanding of the level of swim skill achieved. Future studies should examine the repeated exposure or additional dose in order to better understand the exposure needed to reach swim skill proficiency. 

## 5. Limitations

First, the present study evaluated the efficacy of a program developed by iCan Swim and only included the program participants. There was no control group in the present study, nor the randomization of participants to the intervention. This study is not a randomized control trial, and thus the following threats to the internal and external validity exist: Testing (test-retest improvement), selection bias (i.e., self-section to participate in the program), experimenter effects (due to different instructors), and testing effects (less anxious during the last session of testing). Future randomized controlled trials employing these effects will be necessary to exceed the critical beta value based on the 80% power. The observed beta value for the ADHD effect did not exceed the critical beta value based on the 80% power. These results suggest that this analysis was underpowered due to the relatively small sample of individuals with co-occurring ADHD in the present sample (*n* = 17/86). Therefore, these results would require replication with a larger sample of individuals with ASD and co-occurring ADHD. In the future, a waitlist control randomized design rather than a true control design would be appropriate for future studies, given the small number of programs and large demand for adapted aquatics. 

Second, relative to large-scale clinical or epidemiological studies, this study has a small sample size (*n* = 86 participants). However, this is a moderate to large sample compared to the previous studies evaluating adapted learn-to-swim programs or even other adapted physical activity programs [21]. 

Third, participant diagnosis was based on a parent report, as part of the registration process for the iCan Swim program. There was no additional information regarding each criteria for ASD or ADHD (or any other co-occurring conditions) based on the DSM-5, nor was there information regarding symptom severity. Moreover, there was no additional information regarding medication use. It would be useful for future iCan Swim or adapted swim programs to collect more detailed information regarding ASD and other diagnoses, ASD symptom severity, and medication use. These factors may affect the swim level and behavior during the program. The program asks about communication needs/abilities in their registration material, these data were not available in the dataset for which this secondary data analysis was conducted. However, participants that are non-verbal, minimally verbal or use alternative communication (e.g., sign language, PECS) are not excluded from program participation. Participants with behavioral problems are also not excluded from participation. However, if their behavior is harmful to themselves or others, they may be asked to leave the program. Therefore, it is likely that the program included participants with the full range of ASD severity. Importantly, this study includes the largest sample of children with ASD participating in a learn-to-swim program, and thus provides valuable information on the efficacy of group swim lessons for this population.

Fourth, this study was conducted from a single year of programming from a relatively new program (onset 2015). Future studies are needed to replicate and extend this study with additional years of programming. Fifth, no follow-up swim skill assessment was completed to determine skill retention following program completion. Given the number of participants and the geographic spread of program participants across the US, it would be very difficult for iCan Swim to conduct retention assessments. To date, no studies of adapted swim programs for individuals with ASD have collected retention data. Additional studies are needed to determine the retention of swim skills following adapted swim programs in children with ASD. Sixth, this program is short. Future studies are required to determine the dose of programming (how many sessions), for which a non-swimmer with ASD on average becomes an independent and safe swimmer. 

Seventh, although iCan Swim conducts post-program surveys (via survey monkey) with all the parents and volunteers, there is no question that asks parents about their child’s diagnosis or asks volunteers about the diagnosis of the swimmer with whom they worked. In addition, it is possible that parents had multiple children participate in the program and volunteers may have worked with multiple participants in the program. The survey is completed once by the parents and volunteers, thus it is not possible to separate the responses with respect to each participant. Therefore, it is not possible to determine the unique responses of parents of children with ASD or the volunteers that worked with a child with ASD during the program. In the future, the parent survey would need to be completed for each participant. In addition, a question specifying the participant’s disability would need to be added to determine the satisfaction with the program, specifically for parents of children with ASD. Moreover, the volunteer survey would need to be completed for each participant and a question specifying the diagnosis of the swimmer with whom they worked would also need to be added. 

Finally, although the observed beta values for the Time and Time x Time Out of the Water effects exceeded the critical beta value based on the 80% power, the observed beta value for the ADHD effect did not exceed the critical beta value based on the 80% power. These results suggest that this analysis was underpowered due to the relatively small sample of individuals with co-occurring ADHD in the present sample (*n* = 17/86). Therefore, these results would require replication with a larger sample of individuals with ASD and co-occurring ADHD.

## Figures and Tables

**Figure 1 jcm-10-05557-f001:**
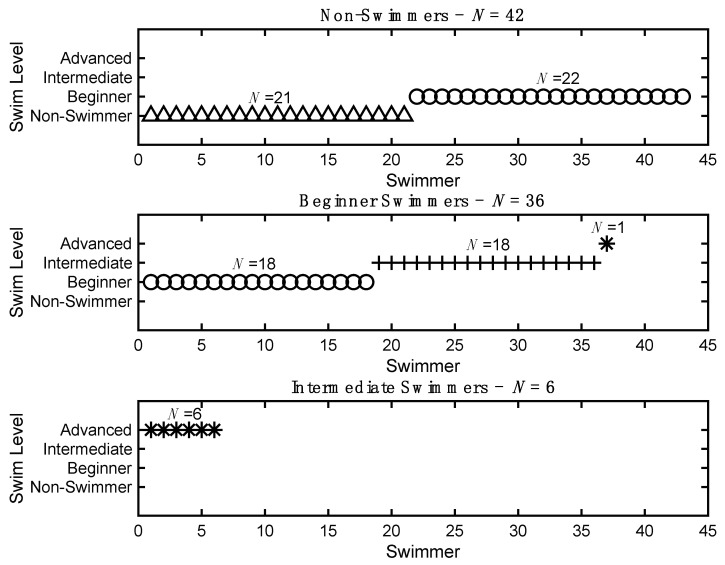
End swim level by start level. (**Top**) participants that started as non-swimmers, (**Middle**) participants that started as beginner swimmers, and (**Bottom**) participants that started as intermediate swimmers. Each participant is depicted as: x = non-swimmer, o = beginner swimmer, + = intermediate swimmer, and * = advanced. The number of participants in each category is indicated. Please note that two advanced swimmers remained at the advanced level at the end of the program, which is not depicted.

**Figure 2 jcm-10-05557-f002:**
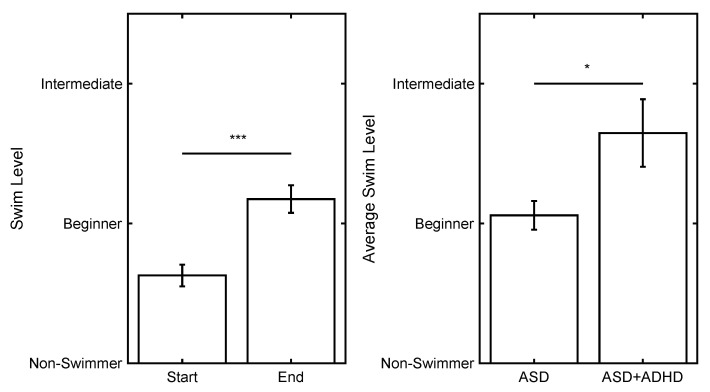
(**Left**) Swim level by time (start/end). (**Right**) Average swim level by ADHD status (ASD/ASD + ADHD). The bars represent the group means, and the error bars represent the standard error of the mean. * = *p* < 0.05, *** = *p* < 0.001.

**Figure 3 jcm-10-05557-f003:**
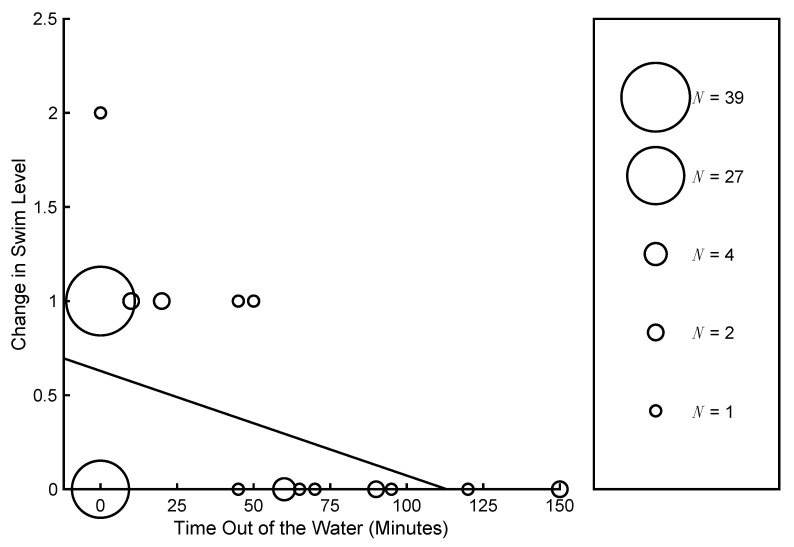
Change in swim level by time out of the water. The size of the circles represents the number of participants with the same change in swim level and time out of the water. The dashed line represents the best linear fit.

**Table 1 jcm-10-05557-t001:** Linear mixed-effects regression estimates, standard error of the estimate, 95% confidence interval, *T*-statistic, degrees of freedom, and *p*-values.

Effect	Estimate	95% Confidence Interval	SE	Estimate(SE)	T(df)	*p*
Intercept	0.29	(−0.17, 0.74)	0.23	0.29(0.23)	1.24(166)	0.22
Age	0.03	(−0.16, 0.08)	0.02	0.03(0.02)	1.3(166)	0.2
ADHD—Yes	0.45	(0.48, 0.84)	0.2	0.45(0.2)	2.21(166)	0.03
Time Out of Water	−0.001	(−0.01, 0.00)	0.003	−0.001(0.003)	−0.49(166)	0.62
Time	0.63	(0.52, 0.74)	0.06	0.63(0.06)	10.98(166)	<0.001
Time Out of Water x Time	−0.006	(−0.01, 0.00)	0.002	−0.006(0.002)	−3.56(166)	<0.001

## Data Availability

The data presented in this study are available on request from the corresponding author.

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
