# Peer review of "Improvements in Swim Skills in Children with Autism Spectrum Disorder Following a 5-Day Adapted Learn-To-Swim Program (iCan Swim)"

_jcm, 2021, doi:10.3390/jcm10235557_

Round 1

Reviewer 1 Report

First, I would like to say that I am very thankful to have the opportunity to read this study. The suggestions given in this document are intended to improve your work. The same feedback document will be given to both editors and authors.

Title: Improvements in swim skills in children with Autism Spectrum Disorder
following a 5-day adapted learn-to-swim program (iCan Swim)
Journal: Journal of Clinical Medicine.

First of all, I would like to congratulate the authors for the research carried out. I understand that this is a relevant topic of study and I agree with the authors that learning to swim is very important for the population of children and adolescents with ASD, not only from the point of view of accident prevention, but also as a healthy physical activity that offers many possibilities for the health and leisure of these populations.

In the article there is a very relevant aspect that has been omitted. There is no explanation of how parental consent was obtained, nor is there any reference to any entity or bioethics commission that would give its prior approval to the study carried out. Bearing in mind that this is a sensitive population and that we are working with minors, this aspect is fundamental for the publication of the article.

On the other hand, although in general terms, the different sections of the work are clear and well structured, I find some aspects that, together, make this study difficult to replicate for future research.For example, I have doubts about the composition of the group of volunteers who participated in the intervention and how they were recruited.For example, I have doubts about the recruitment and composition of the group of volunteers who participated in the intervention and how they were recruited.   It would be very interesting to provide, if necessary as supplementary material, a more detailed description of the interventions developed and the training of both the swimming instructors and the volunteers.

A more specific and detailed description of the criteria for inclusion and exclusion of participants is missing. Furthermore, the researchers do not clarify whether the definitions of ASD and ADHD used in the article refer to the criteria proposed by the DSM-5. Nor do they provide information on the level of severity of symptoms in the case of ASD and whether this level of impairment of the participants was reflected in the initial level of competence in the aquatic environment. It is true that the researchers have noted the latter aspect among the limitations.

Some kind of evaluation of the programme by volunteers, users and/or their parents is also missing. And, finally, whether any post-evaluation was carried out to check whether the effects of the intervention are long-lasting.

Reviewer 2 Report

This short manuscript reports the effects of a swimming program on the swimming skills in children with ASD. Some improvements to the manuscript can be considered:

  1. The STROBE statement should be followed and the relevant checklist can be filled and added in the appendix. The manuscript would require accordingly revision.
  2. Was there a protocol of the study? If yes, please added in the appendix.
  3. The previous study from which data were generated should be briefly discussed and mentioned. This is important also for the waiver about the ethical approval, as it is stated in the relevant section.
  4. The abstract could report effect-sizes, rather than only p-values
  5. The expression in the abstract "…with ADHD and co-occuring ASD…" probably refers to "…with ASD and co-occuring ADHD…"
  6. It should be stated in the limitations that the ASD severity levels were not considered. In addition, concomittant medications (ASDA may be receive sedative psychotrocpics) or comorbidities (that are common in ASD and may affect swimming) were not considered. Were there available data?
  7. It is not clear how the variables were selected for the model, and which process was followed to select interactions or not. Sensitivity analyses can also be considered.
  8. For example, baseline swimming levels seems that should have been in the model, e.g. ADHD participants had also higher baseline swimming levels.
  9. If there are any RCTs they should be discussed.
  10. Since the iCan Swim is a programm of iCan Shine (the second author is the director of iCan Shine), potential conflicts of interest should be more clearly written.
  11. The conclusions should be milder, since this is not an RCT.
